# Untargeted Metabolomics Approach Reveals Diverse Responses of *Pastinaca Sativa* to Ozone and Wounding Stresses

**DOI:** 10.3390/metabo9070153

**Published:** 2019-07-23

**Authors:** Gianni Galati, Anthony Gandin, Yves Jolivet, Romain Larbat, Alain Hehn

**Affiliations:** 1INRA, LAE, Université de Lorraine, 54000 Nancy France; 2Université de Lorraine, AgroParisTech, INRA, UMR Silva, 54000 Nancy, France

**Keywords:** metabolomics, elicitation, parsnip, wounding, ozone

## Abstract

Stresses such as wounding or atmospheric pollutant exposure have a significant impact on plant fitness. Since it has been widely described that the metabolome directly reflects plant physiological status, a way to assess this impact is to perform a global metabolomic analysis. In this study, we investigated the effect of two abiotic stresses (mechanical wounding and ozone exposure) on parsnip metabolic balance using a liquid chromatography-mass spectrometry-based untargeted metabolomic approach. For this purpose, parsnip leaves were submitted to an acute ozone exposure or were mechanically wounded and sampled 24, 48, and 72 h post-treatment. Multivariate and univariate statistical analyses highlighted numerous differentially-accumulated metabolic features as a function of time and treatment. Mechanical wounding led to a more differentiated response than ozone exposure. We found that the levels of coumarins and fatty acyls increased in wounded leaves, while flavonoid concentration decreased in the same conditions. These results provide an overview of metabolic destabilization through differentially-accumulated compounds and provide a better understanding of global plant metabolic changes in defense mechanisms.

## 1. Introduction

During evolution, plants have adapted to stressful environmental conditions by developing multiple survival strategies. The most emblematic one might be the production of lignin, which allows plants to grow erect, facilitates their photosynthesis, and protects them against fungi or insect attacks [1]. However, less visible strategies occurring at different organization levels (molecular, cellular, and physiological) are also triggered by plants. These mechanisms have been extensively described in model plants and crops [2,3]. They are divided into several steps that are initiated by signal perception and the binding of an elicitor to a receptor [4]. In a second step, the signal is conveyed, especially through membrane depolarization and activation of kinase cascades [5]. The third step is characterized by the regulation of the expression of a large set of genes, resulting in the emergence of diverse physiological responses. Some of them are well documented, such as the modulation of photosynthetic activities driven by stomatal closure or metabolic modifications through the accumulation of enzymes and subsequent specialized metabolites [6,7]. These general features have been reported in all plants. Additional specific responses were further described for each plant species and for each kind of stress. This is highlighted by the fine-tuned regulation of the “stress-gene network” orchestrated by the interconnection between the main stress-responsive phytohormones (ethylene, abscisic acid, jasmonic acid, and salicylic acid) [8].

It is well accepted that human activities increase the stressfulness of the environment in the biosphere [9,10]. This stress can be directly associated with the accumulation of atmospheric pollutants or indirectly linked to global climate change. Among atmospheric pollutants, ozone (O_3_) is one of the most studied [11]. It has been shown to impact plant growth and development negatively through the activation of oxidative stress responses and the induction of programmed cell death [12]. To cope with such oxidative stress-related damage and excessive reactive oxygen species (ROS) production, plants have set up a large panel of defense metabolites (e.g., ascorbate-glutathione and polyphenols) or antioxidant enzymes (e.g., catalases and peroxidases) [13]. Since ROS also contributes to cell signaling, it is important for the plant to balance its concentration and avoid excessive accumulation. Yet, the relative contribution of each component is still poorly understood. Other stresses related to global climate change have also been described such as insect herbivory and the modification of the geographic distribution of various pests [14]. The wounds generated by insect attacks stimulate defense mechanisms leading to subsequent modification of the concentration and distribution of primary and specialized metabolites throughout the plant [15,16]. Metabolites are therefore products reflecting cellular functions, and their identification and relative concentration might be considered as a good proxy for assessing the physiological status of plants. The development of high-throughput metabolomic-based methods has already led to the identification and quantification of dozens of metabolites and, in the future, could provide useful insights into the distribution generated by a stimulus on an organism [17,18].

The cultivated parsnip (*Pastinaca sativa*) is an Apiaceae closely related to the carrot (*Daucus carota*), with a white pivotal root acting as a strong carbon sink. This plant is generally reported as a model plant for explaining the coevolution between plant and insect pests. Studies conducted in the 1980s described the emergence and evolution of a metabolic pathway leading to the production of linear and angular furanocoumarins [19]. These molecules are described as phytoalexins displaying phototoxic properties [20]. Upon absorbing UV light, furanocoumarins are able to establish covalent bonds with DNA or proteins, leading to physiological consequences such as blocking cell replication or inactivating enzymes [21]. In recent years, the parsnip has become a model plant for the cloning and functional identification of genes involved in the furanocoumarin biosynthetic pathway [22,23,24,25,26]. However, thus far, little is known regarding the phytochemistry of the parsnip and its response to environmental stresses.

To determine the metabolic response of parsnip leaves to O_3_ and wounding stresses, we designed a liquid chromatography-mass spectrometry (UHPLC-MS) untargeted metabolomic approach. Parsnip plants were subjected to acute O_3_ stress or mechanical wounding (MW). The metabolite composition of the leaves was determined 24 h, 48 h, and 72 h after the initiation of the stresses. The datasets generated were analyzed by multivariate and univariate statistical approaches. Metabolite signals differentially accumulated between treated and control plants were identified and replaced in the corresponding metabolic pathways. MW induced a stronger plant response than did the O_3_ stress. In addition to furanocoumarin accumulation, our study depicted a large modification of the parsnip metabolome in response to MW that included monoterpene accumulation and flavonoid depletion, together with a modulation of the lipid pathway. 

## 2. Material and Methods

### 2.1. Stress Application Settings

Treatments were applied to 2-month-old parsnips grown in six growth chambers (two per treatment) at 21 °C/18 °C day/night and an irradiance of 380 µmol m^−2^ s^−1^ during a 16-h photoperiod. Mechanical wounding was performed on every plant’s leaflets using a device consisting of a regular arrangement of straight iron pins as described by Roselli et al. [24]. For the O_3_ treatment, a preliminary experiment was conducted in order to define the most suitable concentration to apply. Parsnip plants were submitted to ozone treatments (i.e., 0, 80, 120, 150, and 250 ppb, 4 plants/concentration) for 6 h/day over 3 days. No visual symptom was observed for any condition. Therefore, the highest O_3_ concentration was used in the subsequent experiment for the same period as before. For the three conditions (mechanical wounding, O_3_, and control), twenty-four plants were distributed in six different growth chambers (8 plants/treatment and 2 growth chambers/treatment, meaning 4 plants/growth chamber) one week before the application of the stress. The six growth chambers had the same light, temperature, hygrometry, and watering conditions and were under a continuous charcoal-filtered air flux. O_3_ (250 ppb) was added to the air flux of two growth chambers for a period of 6 h/day over 3 days. O_3_ was generated from pure O_2_ with an ozone generator (OZ500; Fischer, Bonn, Germany and CMG3-3; Innovatec II, Rheinbach, Germany). Air flux allowed a rapid air renewal in the growth chamber (<2 min), meaning that the O_3_ concentration was quickly reached and backed down. This was controlled through continuous monitoring with an ozone analyzer (O_3_41M; Environment S.A., Paris, France). 

For each treatment (mechanical wounding, O_3_, and control) a leaflet was collected from the same eight plants immediately before (i.e., T0) and 24, 48, and 72 h (T1, T2, and T3) after the initiation of the experiment. To ensure that there was no link between leaflet age and metabolic content, we designed a preliminary test consisting of measuring the concentration of 12 coumarins and furanocoumarins (umbelliferone, bergapten, xanthotoxin, isopimpinellin, xanthotoxol, psoralen, osthenol, pimpinellin, demethylsuberosin, angelicin, marmesin, isobergapten) in every leaflet of a plant. Leaflets of 4 *P. sativa* plantlets were collected and subjected to metabolite extraction as described below. The selected metabolites were quantified by UHPLC-MS analyses by using a specific standard curve (1, 5, 20, 30 µM). 4-Methylumbelliferone at a concentration of 5 µM was used as an internal standard for mass quantification. The results of this preliminary experiment are available in Appendix A. Briefly, the cumulative concentrations of all the metabolites measured were not significantly different between the five leaflets from the same parsnip stem. Therefore, the leaflets sampled at T0, T1, T2, and T3 correspond to leaflets from the same stem.

### 2.2. Metabolite Extraction

The frozen parsnip leaves were crushed with a mortar and pestle. Eight hundred microliters of 80% methanol solution were mixed with 100 mg of powder independently collected for each sample. After sonication (sweep mode, Elma s70 Elmasonic, Singen, Germany) for 10 min and a 20 min-long centrifugation at 13,000× *g*, the supernatant was transferred into a new 2-mL microtube. A second extraction was performed on the pellet using the same conditions. The two supernatants were pooled and dried at 40 °C using a speed vacuum (Concentrator plus, Eppendorf, Hamburg, Germany). The pellet was suspended in 100 µL of 80% methanol containing 5 µM 4-methyl-umbelliferone as an internal standard.

### 2.3. UHPLC-MS

The plant extracts were analyzed by UHPLC-MS (Prominence, Shimadzu, Kyoto, Japan). The device was equipped with a diode array detector SPDM20A (Shimadzu, Kyoto, Japan) (PDA) and a single quadrupole mass spectrometer LCMS2020 (Shimadzu, Kyoto, Japan). The molecules were separated on a C18 column in inverse phase (ZORBAX Eclipse Plus 150 × 2.10 mm; particle size = 1.8 μm; Agilent Technologies, Santa Clara, CA, United States). The mobile phase consisted of 0.1% formic acid in ultra-pure water (A) and 0.1% formic acid in methanol (B). The molecules were eluted through a gradient as follows: (A:B, v/v), (90:10) at 0 min, (80:20) at 1 min, (40:60) at 6 min, (10:90) at 10 min, (0:100) from 12 min–16 min, and (90:10) from 16 min–20 min. The solvent flow was set at 0.2 mL/min, and the injection volume was set at 5 µL. The compounds were detected using a UV detector (200–600 nm). They were further analyzed with a mass spectrometer using a double ionization source (DUIS) set up in positive mode and by the combination of electrospray ionization and chemical ionization with atmospheric pressure. The electrospray ionization received a tension of 4.5 kV, and the temperatures of the heating block, the entrance, and the line of desolvation of the mass spectrometer were set at 400 °C, 350 °C, and 250 °C, respectively.

### 2.4. High-Resolution Mass Spectrometry

High-resolution mass spectrometry (HRMS) and tandem mass spectrometry (MS/MS) analyses were performed on the selected samples. Forty microliters of the sample were analyzed on an HPLC-MS LTQ-Orbitrap consisting of a liquid chromatography device (Dionex UltiMate^®^ 3000, Thermo Scientific, Waltham, MA, United States) coupled with a quaternary solvent delivery pump and a linear ion trap mass spectrometer (LTQ XL™, Thermo Scientific, Waltham, MA, United States) connected to an Orbitrap HRMS (Thermo Scientific, Waltham, MA, United States). The separation of the compounds was performed on a C18 column in inverse phase (Alltima™ 150 × 2.1 mm; size of particle = 5 µm, Alltech^®^). The mobile phase consisted of 0.1% formic acid in ultra-pure water (A) and 0.1% formic acid in methanol (B). The gradient was set up as follows: (A:B, v/v), (90:10) at 0 min, (43:57) at 20 min, (35:54) at 32 min, (10:90) at 38 min, and (90:10) at 40 min. The flow was set at 0.2 mL/min for a 40-min analysis. The tandem mass spectrometer was connected to a liquid chromatography system and a source of ionization ESI set in positive mode with a tension of 4.5 kV and a temperature of 300 °C. The flows of coaxial, auxiliary gases, and barrier were adjusted to 40 and 10 arbitrary units (AU)/min, respectively. The voltage of the capillary of transfer, split lens, and forehead (front) lens at the entrance of the source of ionization were set at 36 V, 44 V, and 3.5 V. The analysis started from m/z ratios of 50–1000.

### 2.5. ROS Analysis

Hydrogen peroxides were extracted from 100-mg FW of leaves ground in liquid nitrogen and 5% PVPP and suspended in 0.1% TCA. The sample was centrifuged at 18,000× *g* for 15 min at 4 °C. The supernatant was assayed in 5 mM potassium phosphate buffer (pH 7.0) and 0.75 M KI (according to [27]). The absorbance was immediately measured at 390 nm. A Kruskal–Wallis test was performed for statistical analysis.

### 2.6. Metabolic Data Analysis

The UHPLC-MS raw data were converted to .cdf files using the LabSolution software (Shimadzu, Japan) and were pre-processed using the XCMS Online platform (https://xcmsonline.scripps.edu). This pre-processing consisted of peak detection, retention time correction, and alignment of the molecules. Peak detection was performed with the CentWave method (Δm/z = 2.5 ppm, peak width (5–50), Signal-Noise threshold = 20). Retention time correction was performed with the obiwarp method, and alignment was performed with the following parameters (minfrac = 0.5, bw = 5, mzwid = 0.015). An analysis was performed by comparing the eight replicates of each treatment at a given harvest time (T0, T1, T2, and T3). The pre-processed datasets, corresponding to 1592, 1077, 1091, and 1260 features in positive mode at T0, T1, T2, and T3, respectively, were imported into Microsoft Excel and converted to .csv files. Multivariate and univariate statistical analyses were performed with the MetaboAnalyst 4.0 online platform (https://www.metaboanalyst.ca) [28]. Prior to the statistical analyses, datasets were filtered based on the relative standard deviation (RSD/mean) to remove the features with low repeatability (40% of the filtered features). Then, data were normalized using the Pareto method (each feature was mean centered and divided by the square of the standard deviation). A principal component analysis (PCA) was conducted on the filtered and normalized features, corresponding to 956, 641, 655, and 755 features for T0, T1, T2, and T3, respectively. A two-way analysis of variance (ANOVA, *p* < 0.001) was performed to identify metabolic features affected by treatment and sampling date. A one-way analysis of variance (ANOVA, *p* < 0.05, Fisher’s least significant difference (LSD) post hoc test) was performed to identify the metabolic features significantly affected by O_3_ and MW stresses at a specific sampling date. In each case, a false discovery rate approach was performed as multiple testing correction. 

Metabolite identification was performed based on significantly affected metabolic features. Raw HPLC-MS LTQ-Orbitrap data files containing exact mass and mass fragmentation data were converted to .abf and analyzed using MS-Dial (v.3.20) and MS-Finder (v.3.12) software. HRMS data allowed the determination of the molecular formula, while MS/MS data were used to propose molecular structures by comparison with 21 metabolomic databases, including UNPD, LipidMAPS, HMDB, MassBank, METLIN, and ReSpect. When possible, the identification of metabolites was confirmed by comparison with authentic standards. This was the case for rutin, imperatorin, osthole, 8-geranyloxy-psoralen, oleic acid, linolenic acid, and LysoPC (16:0).

## 3. Results

### 3.1. Impact of Treatment on the Oxidative Status of Parsnip Leaves

To characterize the intensity of oxidative stresses, the hydrogen peroxide (H_2_O_2_) concentration was assessed in parsnip leaves that were exposed to O_3_ or mechanically wounded (MW). After three days of O_3_ exposure (250 ppb), parsnip leaves did not show any chlorosis or early senescence symptoms. H_2_O_2_ content was measured for the O_3_, MW, and control plants. Although H_2_O_2_ content tended to be higher for the O_3_ treatment, this observation was not statistically significant (Appendix A).

### 3.2. Global Overview of the Metabolomic Profile.

To determine the global metabolic response in MW- and O_3_-treated parsnip leaves, we developed an untargeted metabolomics strategy. For this purpose, LC-MS analyses were performed on the hydro-methanolic extracts prepared from the leaves collected 24, 48, and 72 h after the beginning of the stresses. Metabolic features, characterized by a mass on charge ratio in positive mode (m/z) and the retention time, were detected and quantified in each sample (Appendix A). In order to assess the impact of treatment on the parsnip leaf metabolome, we performed a PCA on the metabolic features at each sampling date (Figure 1). 

From these PCA, it appears that the treatments, overlapping with the control at T0, evolved differently after T1. MW-treated plants tended to separate from the control at T1 and were clearly distinct from control plants at T2 and T3. On the other hand, O_3_ treatment was mainly overlapped with the control whatever the sampling time.

Metabolic features significantly affected by the treatments and the sampling time were identified by a two-way ANOVA (*p* < 0.001). The heatmap constructed with the significant metabolic features (Figure 2) led to identifying clusters of metabolic features that were positively (Cluster I, Figure 2) or negatively (Cluster III, Figure 2) affected, depending on the sampling time. In addition, three main trends can be visualized regarding the impact of treatments: Cluster II gathered metabolic features that tended to decrease both under O_3_ and MW; Cluster IV contained few metabolic features accumulating under O_3_ in comparison with the control; and Cluster V gathered metabolic features gradually accumulating under MW in comparison to both O_3_ and control samples (Figure 2). 

The number of differentially-accumulated metabolic features (DAMFs) between the two stress treatments (O_3_ and MW) and the control was determined at each sampling date by a one-way ANOVA (*p* < 0.05) (Table 1). This analysis gave evidence that the MW treatment led to a greater modification of the parsnip leaf metabolome than the O_3_ stress. The number of DAMFs was increased at T1. It was maximum at T2 for MW, whereas it decreased after T1 for the O_3_ treatment. MW was characterized by a large majority of DAMFs increasing in abundance, whatever the sampling date. 

### 3.3. Metabolite Identification

Metabolite identification was performed on the DAMFs at T1, T2, and T3, which represented a dataset of 544 non-redundant entities. This dataset was filtered by subtracting fragment ions and metabolic features with a small peak area or with non-exploitable MS/MS data, leading to a final dataset of 245 DAMFs. High-resolution mass spectrometry and MS/MS fragmentation data led to the identification of 39 metabolites (Table 2). The identified DAMFs were mostly related to MW rather than to O_3_ exposure (250 ppb). According to the Classyfire metabolic annotation [29], the putatively-identified metabolites were distributed among eight major classes: flavonoids, fatty acyls, coumarins and derivatives, indoles and derivatives, tetrapyrroles and derivatives, prenol lipids, glycerophospholipids and glycerolipids (Table 2). According to their response to the two stresses, the identified metabolites can be grouped into three categories. The first group (A) contained metabolites whose accumulation was significantly reduced by O_3_, MW, or both stresses. It comprised all the flavonoids identified, several fatty acyls, and glycerolipids. The accumulation of these metabolites was significantly reduced under MW at T2 and T3. On the other hand, only a few of these metabolites were affected by O_3_ at T1 and T2 only. 

The second group (B) included metabolites whose accumulation was massively increased under O_3_ at T1 and T2 and under MW at T2 and T3. One of these metabolites (corresponding to the feature labeled M677T8, Table 2) was identified as a tetrapyrrole derivative based on its chemical formula and fragmentation pattern. Such metabolites, referred to also as phyllobilins or non-fluorescent chlorophyll catabolites (NCCs), were discovered three decades ago in senescent leaves and in maturing fruits [30]. NCCs are characterized by a complex structure and a high diversity, making them difficult to identify. However, recent mass spectrometry approaches contributed to their structural elucidation [31]. Based on these methods and by comparison with the mass fragmentation of M677T8, we assumed that four additional metabolites could be assigned as NCCs (Appendix A). 

The third group (C) contained metabolites, the accumulation of which was exclusively induced under MW treatment. It included compounds related to primary metabolism such as tryptophan, several fatty acids, and glycerophospholipids. As expected, apart from these molecules, specialized metabolites such as monoterpenoids, coumarins, and furanocoumarins were also significantly induced. The accumulation of all these metabolites increased up to T2 and slightly decreased at T3 in most cases. 

## 4. Discussion

Plants have developed several strategies to adapt to various environmental conditions. A way to investigate these strategies consists of applying different stresses to plants and assessing the impact on their phenotype or their metabolome. The O_3_ stress that we applied in our experimental conditions (250 ppb) can be considered as very severe compared to natural conditions (40 ppb on average on Earth) [32]. Such stress is generally reported to induce chlorotic or necrotic symptoms on leaves. In the case of the parsnip, no visible symptoms could be detected even after 72 h. To cope with such stresses, some plant species have developed an “avoiding” strategy and limit O_3_ influx to the leaf tissue by closing their stomata. This was reported in *Amaranthus palmeri* (Amaranthaceae), for example [33]. Such a strategy usually contributes to minimizing oxidative stress and subsequent ROS accumulation. The absence of H_2_O_2_ increase in parsnip leaves after O_3_ treatment (Appendix A) supports this strategy. However, the resistance of the parsnip to oxidative stress may also rely on other alternative strategies such as a large detoxifying arsenal involving specialized metabolites like flavonoids that are well known for their strong antioxidant properties. Such a strategy has already been reported in the poplar for which the synthesis of flavonoids is increased after O_3_ exposure [34]. However, this is not a general rule since the flavonoid concentration was also shown to be mostly reduced in leaves of *Brassica nigra* after O_3_ exposure [35,36]. Concerning O_3_-treated parsnip leaves, the concentration of flavonoids was generally not significantly different from that in the control except for two flavonoids that transiently decreased. O_3_ exposure led also to an increase in NCC. This increase might be the consequence of chlorophyll degradation in response to O_3_ exposure, as already reported in the literature [37,38,39]. The role of NCCs in plants is still under investigation; however, some of them were demonstrated to possess strong antioxidant properties [31]. Their accumulation under O_3_ exposure may thus contribute to parsnip tolerance to this stress through their antioxidant properties. However, these molecules are probably not the only ones involved in this resistance. Additional primary antioxidant molecules (e.g., ascorbate or glutathione) or detoxifying enzymes (e.g., catalase or peroxidase) might have a significant role, and further investigations will be necessary to evaluate their contributions.

MW has widely been used as a tool to simulate plant response to herbivory [40]. This approach was shown to be effective at mimicking qualitatively and sometimes quantitatively the main mechanisms of plant response to herbivores [41,42], even though the addition of insect oral secretions that contain several molecules usually leads to more specific plant responses [43]. In our study, MW significantly affected the metabolic composition of parsnip leaves, as indicated by the accumulation of coumarins and furanocoumarins. This effect has previously been shown in roots of cultivated parsnips [24,25,26] and leaves of wild parsnips submitted to either mechanical wounding or herbivory [44,45]. Interestingly, in these older studies, the panel of induced furanocoumarins was different from the one highlighted in our results. Indeed, we did not observe an induction of the synthesis of simple furanocoumarins, such as psoralen, xanthotoxin, and isopimpinellin, but we noticed a significant increase in prenylated molecules such as 8-geranyloxy-psoralen. To our knowledge, this has not been reported in parsnip leaves so far. 

Concomitantly with the furanocoumarin increase, we observed that the concentration of flavonoids decreased in the parsnip leaves submitted to mechanical wounding. This result is surprising since the flavonoid pathway is generally activated in response to biotic and abiotic stresses [46]. An explanation might be related to the biochemical relationship between the furanocoumarin and the flavonoid pathways. Both specialized metabolite pathways share a common precursor, *p*-coumaroyl-CoA (Figure 3). The induction of the synthesis of the prenylated furanocoumarin upon mechanical wounding may redirect the carbon flux to this pathway at the expense of the biosynthesis of flavonoids. Such metabolic competition for a common precursor has previously been demonstrated in genetically-modified *Arabidopsis thaliana* expressing a bacterial bi-functional chorismate mutase/prephenate dehydratase [47].

MW enhanced the accumulation of two monoterpenoids. This accumulation was significantly different, from the control, one day after the initiation of the stress and reached a maximum after two days. Enhanced accumulation of such molecules in response to MW is well established since they were already reported to be involved in direct and indirect plant defense mechanisms in other plants such as *Myrica cerifera* and *Artemisia annua* [48,49]. 

In addition to specialized metabolites, MW also led to significant modifications in the lipid composition of the leaf extract with a global increase in fatty acyls and glycerophospholipids and a decrease in glycerolipids. Such modifications were previously reported [50], and comprehensive mechanisms involving lipids in the signaling of plant response to abiotic stress have been reviewed. The modifications in the lipid profile occurring in parsnip leaves in response to MW are consistent with these previously-described mechanisms. Hence, the accumulation of lysophospholipids (LysoPC (16:0), LysoPC (18:3), and LysoPC (18:2); Table 2) and fatty acids (stearidonic acid, linolenic acid, and oleic acid; Table 2) may be the consequence of the activation of phospholipases that cleave fatty acids from phospholipids [51]. The free fatty acids that we have detected also displayed a high degree of desaturation, which has previously been associated with plant responses to biotic and abiotic stresses and may be the consequence of the activation of fatty acid desaturases [52]. The accumulation of 13-HOTE, generated through the oxidation of linolenic acid by a lipoxygenase, and the corresponding accumulation of linolenic acid indicate that the oxylipin pathway was activated. This pathway notably leads to the synthesis of jasmonic acid, a well-known phytohormone involved in plant responses to biotic and abiotic stresses [53,54]. The decrease in glycerolipids, namely diacylglycerols and monogalactosyldiacylglycerols, may be the consequence of the activation of phospholipases or may illustrate the degradation of plastids that are rich in glycerolipids. Such a decrease has also been observed in wounded *A. thaliana* [55,56].

In conclusion, our study aimed at obtaining a comprehensive understanding of the parsnip response to O_3_ fumigation and MW using an untargeted metabolomic approach. Despite exposure to a high O_3_ concentration (250 ppb), the oxidative status and the metabolome of the parsnip leaves were only slightly affected in comparison to the control plants. For MW, the impact was much higher. This result is consistent with studies done on furanocoumarins [24,25,57]. In our study, we also highlight a concomitant decrease in the concentration of flavonoids that may be the consequence of a competition between both metabolic pathways sharing a common precursor (i.e., *p*-coumaroyl-CoA). Additional experiments will however be required to validate this hypothesis and understand the underlying regulation mechanisms. MW also enhances the production of terpenes, and the lipid composition was severely affected with an increase in free fatty acyls and glycerophospholipids and a decrease in glycerolipids. This is in accordance with the induction of the oxylipin pathway, leading especially to the defense phytohormone jasmonic acid. This study revealed that this plant displays different strategies in response to two stresses applied individually. It would now be interesting to extend this study by their combination, which might be more related to natural conditions. 

## Figures and Tables

**Figure 1 metabolites-09-00153-f001:**
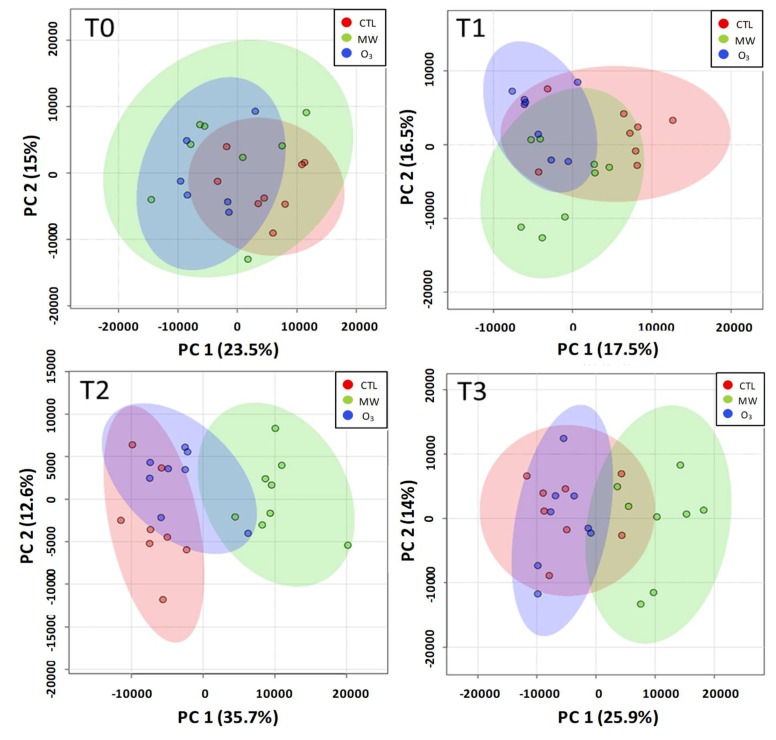
Principal component analysis (PCA) of the parsnip metabolomic datasets. PCA analyses were performed at T0, T1 (24 h), T2 (48 h), and T3 (72 h) after the application of the stresses. The red, green, and blue areas and dots are related to control, mechanical wounding (MW), and ozone stress (250 ppb), respectively. For each PCA, the number of samples was 24 (eight replicates/treatment). The number of variables was 956, 641, 655, and 755 for PCA at T0, T1, T2, and T3, respectively. The number of principal components fitted was set to eight for each PCA. The cumulative variance explained was 76.6%, 77.9%, 83.1%, and 76.8% for PCA at T0, T1, T2, and T3, respectively.

**Figure 2 metabolites-09-00153-f002:**
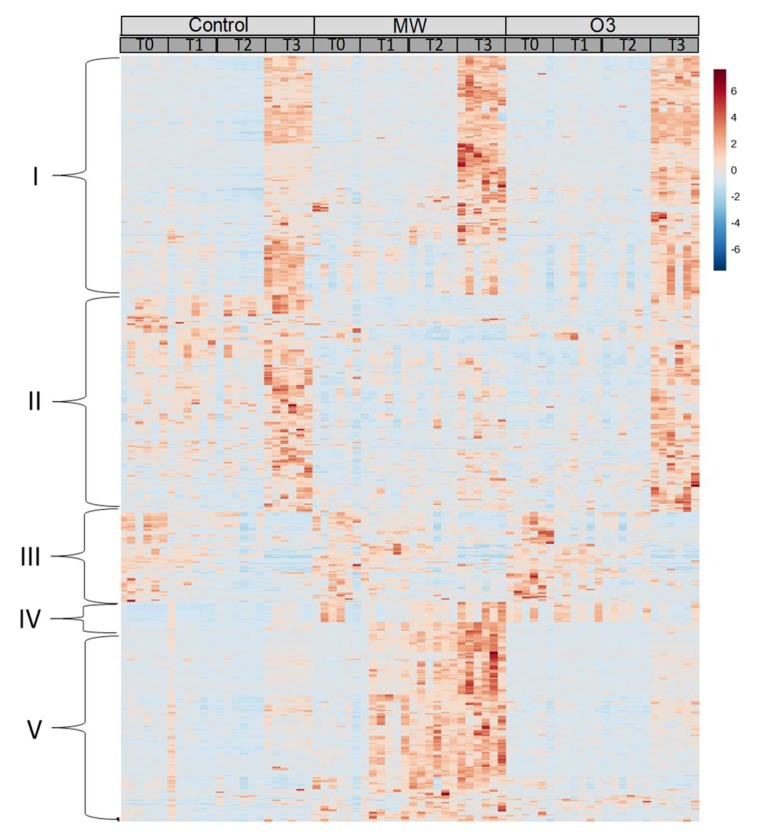
Heatmap plot of the 646 significant metabolic features determined by a two-way ANOVA on time and treatment (*p* < 0.001) at T0, T1 (24 h), T2 (48 h), and T3 (72 h) after the application of the stresses (O_3_ or MW). The red and blue colors indicate a higher and lower intensity level for each metabolic feature, respectively. Each line refers to a single metabolic feature. Each column refers to a sample. Clusters of significant metabolic features are identified by the mark on the left. Each cluster is depicted in the Results Section.

**Figure 3 metabolites-09-00153-f003:**
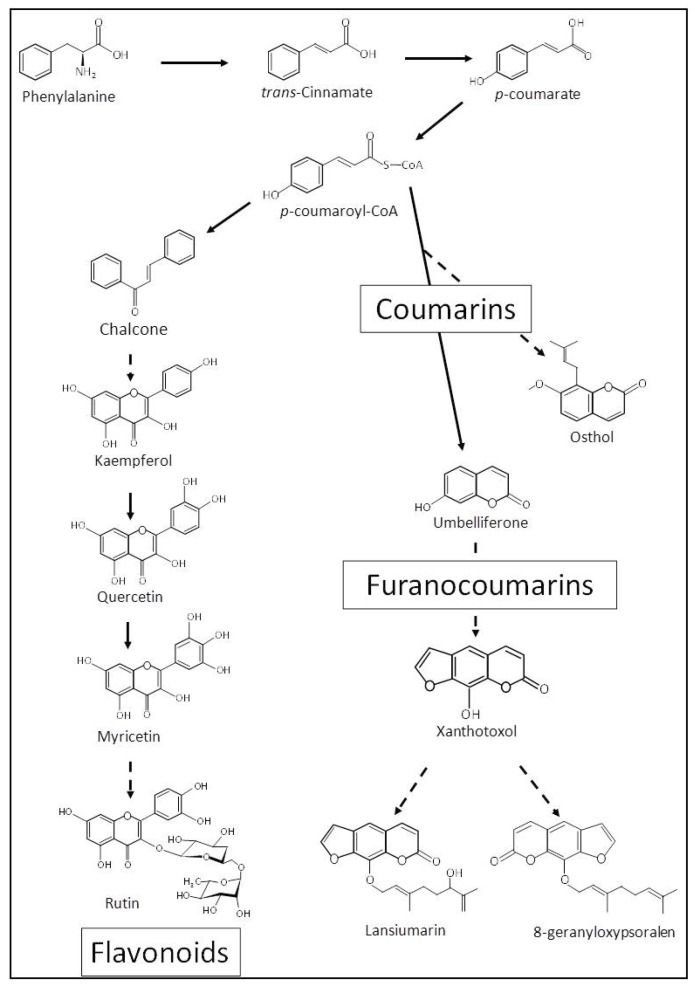
Schematic diagram of putative *p*-coumaroyl-CoA competition between flavonoid and furanocoumarin pathways. Continuous arrows represent a one-step reaction; dotted arrows represent several enzymatic reactions.

**Table 1 metabolites-09-00153-t001:** Number of differentially-accumulated metabolic features (DAMFs) for the O_3_ (250 ppb) and MW treatments in comparison to the control plant. The DAMFs are categorized according to the sampling time (T0–T3), the treatment (O_3_ and MW), and their accumulation (up)/depletion (down). The DAMFs were identified from a one-way analysis of variance (ANOVA, *p* < 0.05, Fisher’s least significant difference (LSD) post hoc test). The detailed list of all metabolic features and DAMFs is available in Appendix A.

	O_3_/Control		MW/Control	
	Down	Up	Down	Up
**T0**	1	1	1	0
**T1**	19	42	17	111
**T2**	19	6	41	304
**T3**	10	16	82	185

**Table 2 metabolites-09-00153-t002:** Identification of the DAMFs. The identified DAMFs are listed and categorized into three groups (A, B, C). Group A contains DAMFs negatively affected by O_3_, MW, or both; Group B contains DAMFs over accumulated under O_3_ (250 ppb) and MW treatment; Group C contains DAMFs over accumulated only under MW treatment. For each DAMF, the fold change (FC) of the treatment/control is given only when significant (*p* < 0.001).

Group	Feature Name	FC O3/Control			FC MW/Control			Ion Adduct	RT (min)	[M + H] ^+^ *m/z*	Molecular Weight	Molecular Formula	Assigned Compound	Classyfire Class
		T1	T2	T3	T1	T2	T3							
A	M293T11	_	_	_	_	_	0.4	[M + Na]^+^	10.56	293.078	270.28	C16H14O4	Imperatorin	Coumarins and derivatives
A	M595T8	_	_	_	_	_	0.7	[M + H]^+^	7.55	595.165	594.52	C27H30O15	Kaempferol 3-rhamnoside-7-glucoside	Flavonoids
A	M611T6	_	_	_	_	_	0.5	[M + H]^+^	6.43	611.160	610.52	C27H30O16	Rutin	Flavonoids
A	M627T5	_	_	_	_	_	0.6	[M + H]^+^	5.09	627.154	626.52	C27H30O17	Quercetin 3, 4-diglucoside	Flavonoids
A	M625T8	0.5	0.6	_	_	_	0.5	[M + H]^+^	7.75	625.178	624.54	C28H32O16	Myricetin 7-methyl ether 3,4’-di-O-alpha-L-rhamnopyranoside	Flavonoids
A	M757T6	_	0.5	_	_	0.7	0.5	[M + H]^+^	6.46	757.218	756.66	C33H40O20	Quercetin-3-O-alpha-L-rhamnopyranosyl(1-2)-beta-D-glucopyranoside-7-O-alpha-L-rhamnopyranoside	Flavonoids
A	M751T13	_	_	_	_	0.4	0.3	[M + Na]^+^	13.37	751.422	728.99	C43H68O9	CID 11104554	Fatty Acyls
A	M585T16	_	0.7	_	_	0.2	0.4	[M + H]^+^	16.01	585.447	584.87	C37H60O5	DG(20:5(5Z,8Z,11Z,14Z,17Z)/14:1(9Z)/0:0)	Glycerolipids
A	M583T15	_	_	_	_	_	0.5	[M + Na]^+^	14.58	583.435	560.84	C35H60O5	DG(14:1(9Z)/18:3(9Z,12Z,15Z)/0:0)	Glycerolipids
A	M769T16	_	_	_	_	0.3	0.4	[M + Na]^+^	16.00	769.482	747.01	C43H70O10	18:3/16:3-MGD	Glycerolipids
A	M783T15	_	_	_	_	0.5	0.4	[M + Na]^+^	14.71	783.460	760.99	C43H68O11	oxy phytodienoic acid/16:3-MGD	Glycerolipids
B	M677T8	3.0	2.2	0.4	_	2.1	1.6	[M + H]^+^	8.43	677.263	676.72	C38H36N4O8	(2S,2(1)R)-2(1),2(2)-dicarboxy-8-ethenyl-2,7,12,18-tetramethyl-2,2(1)-dihydrobenzo[b]porphyrin-13,17-dipropanoic acid	Tetrapyrroles and derivatives
B	M661T9	3.9	2.6	_	_	2.4	_	[M + H]^+^	8.67	661.283	660.76	C42H36N4O4	Putative chlorophyll catabolite	Tetrapyrroles and derivatives
B	M707T9	3.2	2.9	0.4	_	2.7	2.2	[M + H]^+^	9.45	707.293	706.74	C36H42N4O11	Putative chlorophyll catabolite	Tetrapyrroles and derivatives
B	M693T9	3.0	2.4	_	_	2.1	_	[M + H]^+^	9.14	693.277	692.71	C35H40N4O11	Putative chlorophyll catabolite	Tetrapyrroles and derivatives
B	M691T10	2.9	2.5	0.6	_	3.1	2.4	[M + H]^+^	9.52	691.261	690.70	C35H38N4O11	Putative chlorophyll catabolite	Tetrapyrroles and derivatives
C	M135T11	_	_	_	3.2	6.9	4.2	[M + H]^+^	10.64	135.116	134.22	C10H14	p-cymene	Prenol lipids
C	M151T11	_	_	_	2.6	3.9	2.0	[M + H]^+^	10.74	151.111	150.22	C10H14O	Thymol	Prenol lipids
C	M387T6	_	_	_	1.7	2.2	1.9	[M + Na]^+^	5.60	387.066	364.30	C17H16O9	Xanthotoxol glucoside	Coumarins and derivatives
C	M205T4	_	_	_	_	1.8	_	[M + H]^+^	4.20	205.096	204.23	C11H12N2O2	Tryptophan	Indoles and derivatives
C	M197T7	_		_	_		1.8	[M + H]^+^	6.86	197.116	196.24	C11H16O3	4-(3-hydroxybutyl)-2-methoxyphenol	Phenols
C	M267T11	_	_	_	_	3.2	3.1	[M + Na]^+^	10.73	245.116	244.28	C15H16O3	Osthole	Coumarins and derivatives
C	M335T10	_	_	_	_	8.4	6.2	[M + H]^+^	9.57	335.127	334.36	C21H18O4	Anhydronotoptol derivative 1	Coumarins and derivatives
C	M335T12	_	_	_	2.2	10.2	2.5	[M + H]^+^	12.01	335.127	334.36	C21H18O4	Anhydronotoptol derivative 2	Coumarins and derivatives
C	M337T11	_	_	_	2.9	6.3	3.0	[M + H]^+^	10.62	337.143	336.38	C21H20O4	Anhydronotoptol	Coumarins and derivatives
C	M353T10	_	_	_	3.4	17.4	11.0	[M + H]^+^	9.63	353.136	352.38	C21H20O5	Lansiumarin A derivative 1	Coumarins and derivatives
C	M353T11	_	_	_	_	7.9	2.3	[M + H]^+^	10.72	353.138	352.38	C21H20O5	Lansiumarin A derivative 2	Coumarins and derivatives
C	M339T13	_	_	_	_	1.7	2.9	[M + H]^+^	12.68	339.159	338.39	C21H22O4	8-geranyloxy psoralen	Coumarins and derivatives
C	M613T15	_	_	_	_	2.7	6.2	[M + H]^+^	14.66	613.480	612.92	C39H64O5	DG(18:3(9Z,12Z,15Z)/18:3(9Z,12Z,15Z)/0:0)	Glycerolipids
C	M277T12	_	_	_	2.5	4.3	3.3	[M + H]^+^	12.33	277.213	276.41	C18H28O2	Stearidonic acid	Fatty Acyls
C	M279T12	_	_	_	_	6.0	5.7	[M + H]^+^	12.18	279.231	278.43	C18H30O2	Linolenic acid	Fatty Acyls
C	M279T13	_	_	_	_	5.4	_	[M + H]^+^	12.80	279.231	278.43	C18H30O2	Linolenic acid	Fatty Acyls
C	M283T14	_	_	_	3.6	14.3	15.0	[M + H]^+^	13.22	283.262	282.46	C18H34O2	Oleic acid	Fatty Acyls
C	M295T12	_	_	_	3.0	17.5	21.4	[M + H]^+^	11.81	295.226	294.42	C18H30O3	13-HOTE	Fatty Acyls
C	M496T14	_	_	_	3.0	4.8	2.7	[M + H]^+^	14.08	496.338	495.63	C24H50NO7P	LysoPC(16:0)	Glycerophospholipids
C	M518T12	_	_	_	_	_	4.9	[M + H]^+^	11.68	518.320	517.63	C26H48NO7P	LysoPC(18:3)	Glycerophospholipids
C	M534T12	_	_	_	3.3	7.2	5.2	[M + H]^+^	11.66	534.318	533.67	C27H52NO7P	LysoPC(18:2/0:0)	Glycerophospholipids
C	M536T12	_	_	_	4.0	11.1	6.5	[M + H]^+^	11.66	536.334	535.65	C26H50NO8P	PC(16:1(9Z)/2:0)	Glycerophospholipids
C	M791T16	_	_	_	2.0	3.8	2.9	[M + H]^+^	15.53	790.558	790.06	C42H80NO10P	PS(18:0/18:1(9Z))	Glycerophospholipids

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
