# Peer review of "Untargeted Metabolomics Approach Reveals Diverse Responses of Pastinaca Sativa to Ozone and Wounding Stresses"

_metabolites, 2019, doi:10.3390/metabo9070153_

Reviewer 1 Report

The manuscript describes the experimental study of Pastinaca sativa (wild parsnip) using non-targeted metabolomics to investigate metabolic features that change with mechanical wounding or ozone exposure. The metabolomics methods used by the research team are sound. However, some data have been misinterpreted. The addition of ozone stress in the design of this study may have made this paper of interest. However, the authors have not provided convincing evidence that the ozone treatment had the intended stress effect. In Figure 1, where ROS results (H2O2) are shown, the results are reported as not being significant. However, the authors report on line 268 that they observed an increase in ROS concentration. This interpretation is incorrect, the statistical results indicate no significant change in ROS concentration. As such, the "avoiding" strategy that the authors describe on line 265 seems the most likely explanation of results found in this investigation. If stomata closed in response to ozone exposure, ROS accumulation and oxidative stress would be minimal (as seen in Figure 1). Presence of an "avoiding" strategy is also supported by the lack of separation of the ozone group from controls in the PCA and the minimual number of differentially abundance metabolic features in the ozone group. Since no multiple testing correction was applied, it's possible that the features that were identified as different in the ozone group were simply false positives.

The differences in the furanocoumarins in the mechanical wounding group have been previously described and therefore do not further the understanding of the 'phytochemistry of parsnip and its response to environmental stress' (lines 72-73) that was the stated goal of the research.  These data do confirm what was learned by Zangerl et al in 1998 when they explored metabolic features that change with mechanical wounding in wild parsnip using GC-MS (the current study utilizes the U-HPLC-MS platform), but novel discoveries from this research were not apparent.

The paper is overall easy to understand. However, if accepted, this paper will require careful review of vocabulary to ensure that the correct terminology is used in all cases. For example, the word chosen in the following places does not seem to be appropriately used: evolution (best to omit, line 182); downregulation (replace with increase/decrease, line 218), over-accumulating (replace with increasing in abundance, line 224); optimum (replace with maximum, line 209) and others. Furthermore, the manuscript would benefit from a more careful description of exactly which groups are being compared when reporting statistical results and application of multiple testing corrections as appropriate. The final paragraph should be rewritten as it is difficult to understand and does not accurately capture the conclusions of the research study.

Author Response

We thank reviewer for his constructive remarks and critics. Here is a point by point answer:

Reviewer 1: The manuscript describes the experimental study of Pastinaca sativa (wild parsnip) using non-targeted metabolomics to investigate metabolic features that change with mechanical wounding or ozone exposure. The metabolomics methods used by the research team are sound. However, some data have been misinterpreted. The addition of ozone stress in the design of this study may have made this paper of interest. However, the authors have not provided convincing evidence that the ozone treatment had the intended stress effect. In Figure 1, where ROS results (H2O2) are shown, the results are reported as not being significant. However, the authors report on line 268 that they observed an increase in ROS concentration. This interpretation is incorrect, the statistical results indicate no significant change in ROS concentration. As such, the "avoiding" strategy that the authors describe on line 265 seems the most likely explanation of results found in this investigation. If stomata closed in response to ozone exposure, ROS accumulation and oxidative stress would be minimal (as seen in Figure 1). Presence of an "avoiding" strategy is also supported by the lack of separation of the ozone group from controls in the PCA and the minimual number of differentially abundance metabolic features in the ozone group. Since no multiple testing correction was applied, it's possible that the features that were identified as different in the ozone group were simply false positives.

Author: We agree with reviewer 1. The H2O2 data are not significantly different between treatments, which is an argument in favor of the “avoiding strategy”. We have modified the sentence line 268. In addition, and following the advice of reviewer 4, the figure 1 has been removed and transformed to a supplementary data (Supplementary data 2)

Reviewer 1: The differences in the furanocoumarins in the mechanical wounding group have been previously described and therefore do not further the understanding of the 'phytochemistry of parsnip and its response to environmental stress' (lines 72-73) that was the stated goal of the research.  These data do confirm what was learned by Zangerl et al in 1998 when they explored metabolic features that change with mechanical wounding in wild parsnip using GC-MS (the current study utilizes the U-HPLC-MS platform), but novel discoveries from this research were not apparent.

Author:  We thank reviewer 1 for this comment which makes evidence that we did not sufficiently highlight the novelties of our study, especially in comparison with Zangerl et al., 1998. Indeed, their study, focused on several metabolites analysed by GC-MS, highlights a parsnip response involving an almost unique increase in furanocoumarin content. Our untargeted metabolomics study gives a much more complete picture of parsnip response to mechanical wounding. Obviously it includes a furanocoumarin increase, but also modification in the content of other specialized metabolites such as terpens (increase) and flavonoids (decrease). Under stressing conditions, the flavonoid decrease is quite unusual. We propose that it is the consequence of a precursor competition between the furanocoumarin and the flavonoid pathway. Finally, our untargeted metabolomics approach gave a quite clear picture of the large modification in the lipid pathways, involving an increase in free fatty acyls and glycerophospholipids and a decrease in glycerolipids. These modifications are comprehensively explained and are suggesting notably the activation of the oxylipin pathway leading to methyl jasmonate. In accordance to this answer, we have modified the conclusion of the paper in order to highlight the novelties of our study.

Reviewer 1: The paper is overall easy to understand. However, if accepted, this paper will require careful review of vocabulary to ensure that the correct terminology is used in all cases. For example, the word chosen in the following places does not seem to be appropriately used: evolution (best to omit, line 182), downregulation (replace with increase/decrease, line 218), over-accumulating (replace with increasing in abundance, line 224), optimum (replace with maximum, line 309) and others

Author: Changes have been made according to the suggestion

Reviewer 1:Furthermore, the manuscript would benefit from a more careful description of exactly which groups are being compared when reporting statistical results and application of multiple testing corrections as appropriate.

Author: Precisions have been added in the results section when reporting on statistical aspects. False Discovery Rate was applied as a multiple testing corrections for all the ANOVA realized. This information has been added in the material and method section.

Reviewer 1:The final paragraph should be rewritten as it is difficult to understand and does not accurately capture the conclusions of the research study.

Author: The last paragraph has been deleted and replaced by a conclusion highlighting the novelties of this study

Reviewer 2 Report

In this manscript, the authors report on a rather small experiment with 24 plants divided over 3 treatments: controls, ozone treatment and mechanical wounding. They harvest leaves of these plants before and at three time points after the start of the treatments. Based on a LC-based metabolomics - including MS/MS and and high resolution MS, they identify groups of compounds that respond differently to the treatments. The changes in specific metabolites are interpreted in the light of their ecological functions.

I liked the quality of the metabolomics procedures and the clarity of the writing: clear structure and short sentences. However, I have some rather serious comments on the experimental set-up. First, the plants were sampled multiple times. On the one hand, a repeated measures analyses is nice, as you can control for individual differences among plants. On the other hand, it might mean that different leaves - of different ages- were sampled over time. The metabolome of younger leaves is quite different from that of older leaves -usually they have higher concentrations of defence metabolites. This may explain why T3 generally shows a higher intensity of signals (Figure 2), also in controls (if this were younger leaves that were harvested). They also may respond differently, especially to wounding. Younger leaves are known to be much more responsive in general, than older leaves.  So differences over time - and possibly differences in responses - may be related to the design. Second, the control group may not be the proper one for the ozone treatment. I assumed that this treatment took place in a different climate room. These plants have been transferred every day for the treatment back and forth. The other climate room may have had different light conditions -even if they have the same lamps et cetera. Also the handling may have affected the metabolome. Assuming that the wounded plants and the  controls were not moved, the control may not have been appropriate for the ozone treated plants. It is not possible to derive from the text what happened. Lastly, how were the leaves harvested? Cutting of a leaf also inflicts a mechanical wound.

I am also missing some other experimental details, eg. how was the wounding applied? I would like to see the justification for the use of 250 ppm ozone, including the pre-test, in the materials section.

I also do not always agree with the interpretation. In line 196 you state that the effect of the MW is transient. I still see quite some separation in figure 2 at T3.

I found it confusion that the numbers of the clusters and the numbers of the groups in the tables are not indicative of similar response dynamics. I suggest to change the latter to A, B, C

Line 331- conclusion: Yes, plant responses are complex and involve many compounds, but that is not really novel.

Figure 1: y-axis values: change comma to decimal point. Why did you use Kruskall-Wallis? Data seem to be rather normally distributed (or could be after transformation). a parametric test would likely show a significant difference. Because you do not have this at the moment, one cannot simply state that the ozone caused ROS to increase, as long as the difference is not significant (line 267-268).

Overall, I was missing some deeper discussion of how the responses overlap and why. There is little critical reflection if and how MW compares to real feeding - or application of herbivore spit to mimic induced responses more realisticallyMS,

Author Response

We thank reviewer for his constructive remarks and critics. Here is a point by point answer:

Reviewer 2: In this manscript, the authors report on a rather small experiment with 24 plants divided over 3 treatments: controls, ozone treatment and mechanical wounding. They harvest leaves of these plants before and at three time points after the start of the treatments. Based on a LC-based metabolomics - including MS/MS and and high resolution MS, they identify groups of compounds that respond differently to the treatments. The changes in specific metabolites are interpreted in the light of their ecological functions.I liked the quality of the metabolomics procedures and the clarity of the writing: clear structure and short sentences. However, I have some rather serious comments on the experimental set-up. First, the plants were sampled multiple times. On the one hand, a repeated measures analyses is nice, as you can control for individual differences among plants. On the other hand, it might mean that different leaves - of different ages- were sampled over time. The metabolome of younger leaves is quite different from that of older leaves -usually they have higher concentrations of defence metabolites. This may explain why T3 generally shows a higher intensity of signals (Figure 2), also in controls (if this were younger leaves that were harvested). They also may respond differently, especially to wounding. Younger leaves are known to be much more responsive in general, than older leaves.  So differences over time - and possibly differences in responses - may be related to the design.

Authors: We agree with reviewer 2 on the advantages and draw-back for multiple samplings on a same plant. Regarding the impact of leaflet age on metabolic content, we have set up a preliminary experiment specifically devoted to this point. On that experiment, we have measured the concentration of 12 metabolites (coumarins and furanocoumarins) on all the leaflets of 4 plants. This experiment led to the conclusion that leaflet age did not impact the content of these metabolites. We now have put the results of this experiment in supplemental data (supplementary data 1). In addition during sampling of the main experiment, we have been extremely attentive to collect the same leaflet for each treatment at a given sampling time.   

Reviewer 2: Second, the control group may not be the proper one for the ozone treatment. I assumed that this treatment took place in a different climate room. These plants have been transferred every day for the treatment back and forth. The other climate room may have had different light conditions -even if they have the same lamps et cetera. Also the handling may have affected the metabolome. Assuming that the wounded plants and the  controls were not moved, the control may not have been appropriate for the ozone treated plants. It is not possible to derive from the text what happened. Lastly, how were the leaves harvested? Cutting of a leaf also inflicts a mechanical wound.

Authors: The materials and methods section from the previous version of the paper was not enough clear on the experimental set up. We now have fixed it. The eight plants used for each treatment were grown in three separated growth chambers sharing the same light, temperature, humidity and watering conditions. Plants were placed in the three different growth chambers 1 week before the initiation of the stress. The PCA at T0 showing no separation of the treatment is illustrative of the similarity of the growing conditions in the three growth chambers. All the growth chambers were under a charcoal-filtered air. For the O3 treatment, O3 was added to the air flux 6h/day for 3 days. O3 was generated from pure O2 with an ozone generator. Air flux allow a rapid air renewal in the growth chamber (< 2min) meaning that the O3 fumigation is quickly and efficiently stopped. This has been assessed through a continuous monitoring with an ozone analyser.

Reviewer 2: I am also missing some other experimental details, eg. how was the wounding applied? I would like to see the justification for the use of 250 ppm ozone, including the pre-test, in the materials section.

Authors: Information on the wounding application have been added. The justification for the use of 250 ppb ozone has been moved to the material and method section.

Reviewer 2: I also do not always agree with the interpretation. In line 196 you state that the effect of the MW is transient. I still see quite some separation in figure 2 at T3.

Authors:  We agree with reviewer 2. The use of the term “transient” is not adapted, since Control and MW are still quite separated at T3. We replaced this term by the sentence “Comparison of the PCA overtime suggested a maximum metabolic response at T2 for both treatments with a remaining impact at T3 for the MW.”

Reviewer 2: I found it confusion that the numbers of the clusters and the numbers of the groups in the tables are not indicative of similar response dynamics. I suggest to change the latter to A, B, C

Authors: We have followed your suggestion

Reviewer 2: Line 331- conclusion: Yes, plant responses are complex and involve many compounds, but that is not really novel.

Authors:  The last paragraph has been deleted and replaced by a conclusion highlighting the novelties of this study

Reviewer 2: Figure 1: y-axis values: change comma to decimal point. Why did you use Kruskall-Wallis? Data seem to be rather normally distributed (or could be after transformation). a parametric test would likely show a significant difference. Because you do not have this at the moment, one cannot simply state that the ozone caused ROS to increase, as long as the difference is not significant (line 267-268).

Authors:   The Kruskall-Wallis test was used since a Shapiro test led to the conclusion that our data did not follow a Gaussian curve. The same conclusion has been obtained for the Log-transformed data. According to the result of the Kruskall-Wallis test, there is no difference for H2O2 content between treatments and the sentence on line 267-268 has been corrected.

Reviewer 2: Overall, I was missing some deeper discussion of how the responses overlap and why. There is little critical reflection if and how MW compares to real feeding - or application of herbivore spit to mimic induced responses more realisticallyMS,

Authors:   Elements have been added in the discussion part

Reviewer 3 Report

This manuscript provides a metabolomic approach to study the responses of P. sativa to wounding stress and ozone exposure. The results provide global profile of plant metabolic changes in response to these stressors. Overall this is an interesting study and provides fundamental knowledge at metabolite level in plants.

Some minor comments to be considered are mentioned below:

·      Lines 80 - Briefly mention in a line or two what your results explain or conclude

·      Line 92 - Delete ‘an’ before 80 %

·      Line 169 -170 – This should be mentioned in the methods section

·      Line 173 – 174 – Rephrase the last sentence

·      Figure 1 – Edit the scale units and place the y- axis label next to the scale. Reduce the size of the figure.

·      Line 188 - Rephrase the last sentence. Is it ‘performed’ instead of ‘realized’?

The scientific names of organisms are not italicized in numerous places of the references section (Reference no. 6, 20, 26, 36, 40, 41, 44). Please double-check them!

Author Response

We thank reviewer for his constructive remarks and critics. Here is a point by point answer:

Reviewer 3: This manuscript provides a metabolomic approach to study the responses of P. sativa to wounding stress and ozone exposure. The results provide global profile of plant metabolic changes in response to these stressors. Overall this is an interesting study and provides fundamental knowledge at metabolite level in plants.

Some minor comments to be considered are mentioned below:

Reviewer 3:      Lines 80 - Briefly mention in a line or two what your results explain or conclude

Authors: The main results are now described at the end of the introduction

Reviewer 3:      Line 92 - Delete ‘an’ before 80 %

Authors: OK

Reviewer 3:     Line 169 -170 – This should be mentioned in the methods section

Authors:  Modification has been done

Reviewer 3:      Line 173 – 174 – Rephrase the last sentence

Authors:  The last sentence has been rephrase to “Although H2O2 content tends to be higher in the O3 treatment, this observation was not statistically significant.”

 Reviewer 3:      Figure 1 – Edit the scale units and place the y- axis label next to the scale. Reduce the size of the figure.

Authors:  Edition on the graph has been done. The Figure 1 has been transformed to a Supplementary data since it conclude to no significant effect.

Reviewer 3:      Line 188 - Rephrase the last sentence. Is it ‘performed’ instead of ‘realized’?

Authors:  We changed to “performed”

Reviewer 3: The scientific names of organisms are not italicized in numerous places of the references section (Reference no. 6, 20, 26, 36, 40, 41, 44). Please double-check them!

Authors:  Double checking has been done

Reviewer 4 Report

The results and the overall metabolomics workflow are performed using appropriate standard protocols of UHPLC-MS and HRMS Orbitrap, processing features with MS-Dial + MS-Finder software and XCMS package, and finally MetaboAnalyst for statistical interpretation.

Authors could cite the Xia & Wishart 2011 - Nature protocol for MetaboAnalyst: https://www.nature.com/articles/nprot.2011.319 Authors should report in the Table which metabolite were identified from which of the several database mentioned in the method section (i.e. UNPD, LipidMAPS, HMDB, MassBank, METLIN and ReSpect). Introduction should describe more in detail the reasons for chosing wound (biotic) and ozone (abiotic) stressors, and the importance of multiple stress responses on plant metabolic systems. See for instance, Suzuki et al 2014 - New Phytologist
Overall, both introduction and discussion need much more extensive reference background. Many sentences are left without source. Just as an example, see the sentences L 46-48: ---- In the methods, authors need to clarify in detail how the O3 treatment was performed (growth-chambers?) as well as how the Mechanical wounding (MW) was applied (scissors? scalpel, on how many leaves?).

In the results: Figure 1 (ROS) could be removed as it take quite some space although the results are not significant.  Regarding the multivariate analysis, there is a large overlap between treatment groups, so I would suggest to the authors to perform a supervised PLS-DA in addition to the PCA (both methods are available in MetaboAnalyst). From the PLS-DA, authors can select and show metabolite VIP scores, which indicate the importance of these variables in the model (also available in MetaboAnalyst tools). This would help the interpretation of the result, in support of their conclusions. Moreover, in all the figure legends authors should mention more clearly the treatments and the ozone regime applied (i.e. 250 ppb).

For the discussion, please refer to this previous study, Papazian et al. 2016 - Plant Physiology http://www.plantphysiol.org/content/172/3/2057  which was performed on a similar system (ozone + caterpillar wounding) combining metabolomics (GC-MS and LC-MS) with transcriptomics. It would be interesting to see the authors discussing their results within this context (especially regarding effects on secondary specialized metabolism, i.e. flavoinoids that they also measured). Moreover, phenotype of exposure to ozone showed no symptoms after 5 days (at 70 ppb), but were clearly visible after 2 weeks long-exposure. Authors can discuss the sensitivity of parsnip to the different ozone levels tested (0-250 ppb) and why they selected the highest concentration if no symptoms were found in any treatments.

Author Response

We thank reviewer for his constructive remarks and critics. Here is a point by point answer:

Reviewer 4:
The results and the overall metabolomics workflow are performed using appropriate standard protocols of UHPLC-MS and HRMS Orbitrap, processing features with MS-Dial + MS-Finder software and XCMS package, and finally MetaboAnalyst for statistical interpretation.  Authors could cite the Xia & Wishart 2011 - Nature protocol for MetaboAnalyst: https://www.nature.com/articles/nprot.2011.31

Authors:Citation has been added

Reviewer 4:

Authors should report in the Table which metabolite were identified from which of the several database mentioned in the method section (i.e. UNPD, LipidMAPS, HMDB, MassBank, METLIN and ReSpect).

Authors:   Compound identification has been mostly realized through the MS-Finder software which used 21 databases including the databases mentioned in the method section (+ METLIN and Respect that were manually checked). Since a majority of the identified metabolites are redundant in several databases we consider that indicating all the databases containing each metabolite will make difficult the reading of the table. Instead we prefer to add the InChikey identifier in the supplemental data.

Reviewer 4:

Introduction should describe more in detail the reasons for chosing wound (biotic) and ozone (abiotic) stressors, and the importance of multiple stress responses on plant metabolic systems. See for instance, Suzuki et al 2014 - New Phytologist

Authors:  This study was not designed to address the question of multiple stress response. For this reason this interesting research area is not mention in the introduction section. However, since this questions are of great interest we have mention them as perspective of this study in the conclusion part.

Reviewer 4:

Overall, both introduction and discussion need much more extensive reference background. Many sentences are left without source. Just as an example, see the sentences L 46-48.

Authors:  Several references have been added in the introduction and discussion

Reviewer 4:

In the methods, authors need to clarify in detail how the O3 treatment was performed (growthchambers?) as well as how the Mechanical wounding (MW) was applied (scissors? scalpel, on how many leaves?)

Authors: Information on the wounding application and the O3 treatment have been added. The justification for the use of 250 ppb ozone has been moved to the material and method section.

Reviewer 4:

In the results: Figure 1 (ROS) could be removed as it take quite some space although the results are not significant.

Authors:  We agree with reviewer 4 to remove the figure 1 and assigned it as a supplementary data.

Reviewer 4:

Regarding the multivariate analysis, there is a large overlap between treatment groups, so I would suggest to the authors to perform a supervised PLSDA in addition to the PCA (both methods are available in MetaboAnalyst). From the PLSDA, authors can select and show metabolite VIP scores, which indicate the importance of these variables in the model (also available in MetaboAnalyst tools). This would help the interpretation of the result, in support of their conclusions.

Authors:  PLS-DA analysis were performed but it appears that Q2 value were quite small (between 0.1 and 0.4) which make us to consider this models not optimal. For this reason we decide to maintain the actual data representation with PCA and ANOVA statistics.

Reviewer 4:

Moreover, in all the figure legends authors should mention more clearly the treatments and the ozone regime applied (i.e. 250 ppb)

Authors: Modifications have been made to enhance the clarity of the figures.

Reviewer 4:

For the discussion, please refer to this previous study, Papazian et al. 2016 - Plant Physiology http://www.plantphysiol.org/content/172/3/2057  which was performed on a similar system (ozone + caterpillar wounding) combining metabolomics (GC-MS and LC-MS) with transcriptomics. It would be interesting to see the authors discussing their results within this context (especially regarding effects on secondary specialized metabolism, i.e. flavoinoids that they also measured). Moreover, phenotype of exposure to ozone showed no symptoms after 5 days (at 70 ppb), but were clearly visible after 2 weeks long-exposure. Authors can discuss the sensitivity of parsnip to the different ozone levels tested (0-250 ppb) and why they selected the highest concentration if no symptoms were found in any treatments.

Authors: A mention to the study of Khaling and coworker (2015) has been added in the discussion part regarding the impact of ozone exposure and secondary metabolites. The low sensitivity of parsnip to high ozone concentration is already discussed in the first paragraph of the discussion section. The presence of symptoms on leaves after O3 fumigation is not indicative of an oxidative stress but rather that plant does not support this stress anymore. Metabolomics analysis appears much more informative on this pre-symptomatic phase in order to understand how the plant copes with the oxidative stress. Since, the parsnip plants had no symptom with the highest O3 concentration tested, and since this very high concentration was expected to led to a high oxidative stress in planta, this latter concentration was chosen.   

Round  2

Reviewer 1 Report

The authors have addressed my major concerns. Now that authors have highlighted which results of this research are novel, I see that some findings may be of value to readers of this journal. The research is primarily a descriptive study, which provides lower impact to readers. However, I think their putative hypothesis that of p-coumaroyl-CoA competition between flavonoid and furanocoumarin pathways is interesting and valuable for future research directions. However, authors neglect to mention testing of this hypothesis as a future direction of this work. I think that validating whether this pathway competition is actually occurring should be mentioned in the conclusion paragraph. In my opinion, that research would be more valuable than looking at the combination of MW and O3 stresses.

The conclusion paragraph is improved but still doesn't explain the most important results as clearly as it needs to. I would direct authors to their response to my 2nd reviewer comment above regarding the novelty of their research beyond furanocoumarins. The response to my comment is much clearer than the conclusion paragraph in the manuscript. For example, on line 364 authors mention a trade-off between the 'two pathways' but do not state that they are referring to the flavonoid and furanocoumarin pathways. I believe that a primary reason I did not understand the full impact of the author's work is that a clarity of language is absent from the manuscript. The research and results are sound, but this manuscript will require substantial, thorough, and careful editing for clarity.

Therefore, I recommend that this manuscript receive extensive editing for English language and style. For example, at a glance, I found at least 3 typos in the new conclusion paragraph alone. I do not have the time to provide this extensive review but I highly recommend that the authors or the journal do so prior to publish. 

Also, authors need to check the headers in file Supdata 1 as some are reading #NAME? instead of the correct header. 

Author Response

Authors: We thanks reviewer 1 for the time spent on this paper and the remarks he addressed to improve it.

Reviewer 1: The authors have addressed my major concerns. Now that authors have highlighted which results of this research are novel, I see that some findings may be of value to readers of this journal. The research is primarily a descriptive study, which provides lower impact to readers. However, I think their putative hypothesis that of p-coumaroyl-CoA competition between flavonoid and furanocoumarin pathways is interesting and valuable for future research directions. However, authors neglect to mention testing of this hypothesis as a future direction of this work. I think that validating whether this pathway competition is actually occurring should be mentioned in the conclusion paragraph. In my opinion, that research would be more valuable than looking at the combination of MW and O3 stresses.

Authors:  We agree with reviewer’s comment. The conclusion has been modified by including the test of the competitive hypothesis as a future prospect.

Reviewer 1: The conclusion paragraph is improved but still doesn't explain the most important results as clearly as it needs to. I would direct authors to their response to my 2nd reviewer comment above regarding the novelty of their research beyond furanocoumarins. The response to my comment is much clearer than the conclusion paragraph in the manuscript. For example, on line 364 authors mention a trade-off between the 'two pathways' but do not state that they are referring to the flavonoid and furanocoumarin pathways. I believe that a primary reason I did not understand the full impact of the author's work is that a clarity of language is absent from the manuscript. The research and results are sound, but this manuscript will require substantial, thorough, and careful editing for clarity.

Therefore, I recommend that this manuscript receive extensive editing for English language and style. For example, at a glance, I found at least 3 typos in the new conclusion paragraph alone. I do not have the time to provide this extensive review but I highly recommend that the authors or the journal do so prior to publish. 

Authors:  The first version of the paper has been extensively reviewed by two editors of American Journal of Expert (Certificate Verification Key: B267-D120-1AE2-524D-7A3E). However, the modifications made during the reviewing process have not been reviewed which resulted in several mistakes. We apologies for this and made extensive editing with native english speakers in order to correct all (or most) of them.

Reviewer 1: Also, authors need to check the headers in file Supdata 1 as some are reading #NAME? instead of the correct header. 

Authors:  The header of the sup data 3 have been modified

Reviewer 2 Report

I appreciate the corrections of the authors. However, doing three different treatments in three different cabinets, without repeating the whole experiment at least once with the same treatements assigned to different cabinets, is not as it should be, even if the metabolomes are overlapping in the first PCA at T0. In ecological sciences, this would be a fatal flaw in the experimental design - the 8 plants in one cabinet would be considered pseudoreplicates. I cannot make it nicer than it is! If this would be a preliminary experiment to a larger one, it would be okay. But as an experiment on itself, not.

As for the differences among the leaflets: are the error bars standard errors or standard errors of the mean? It is hard to follow through the statstical analyses if you only give names of the models, but no details as F values, degrees of freedom et cetera. Moreover, an ANOVA may not be appropriate as the leaflets are interconnected and thus the data are dependent. I also do not get the remark that the A5 leaf was apparently discarded?

The english in the new texts needs to be corrected, it is not as good as the rest.

Author Response

Authors:   We thanks reviewer 2 for the time spent on this paper and the remarks he addressed to improve it.

Reviewer 2: I appreciate the corrections of the authors. However, doing three different treatments in three different cabinets, without repeating the whole experiment at least once with the same treatements assigned to different cabinets, is not as it should be, even if the metabolomes are overlapping in the first PCA at T0. In ecological sciences, this would be a fatal flaw in the experimental design - the 8 plants in one cabinet would be considered pseudoreplicates. I cannot make it nicer than it is! If this would be a preliminary experiment to a larger one, it would be okay. But as an experiment on itself, not.

Authors:   We understand the point of reviewer 2. In fact the three treatments have not been made in three different cabinets by in six. Indeed, we used 2 cabinets/treatment, meaning 4 plants/cabinet. This strategie was used to assess for a potential cabinet effect. We apologise for the mistake made in the description of the method in the revised form of the paper. We have corrected the method section in accordance. The similarity of growth conditions between the six cabinets were monitored to assess there was no difference. The data can be provided if necessary.

Reviewer 2: As for the differences among the leaflets: are the error bars standard errors or standard errors of the mean?

Authors:   The error bars are standards error.

Reviewer 2: It is hard to follow through the statstical analyses if you only give names of the models, but no details as F values, degrees of freedom et cetera.

Authors:   The details of statistical analyses were added

Reviewer 2: Moreover, an ANOVA may not be appropriate as the leaflets are interconnected and thus the data are dependent.

Authors:   According to your remark, we used the same model by integrating individual dependency by adding plant individual as covariable. The results of this model indicate there was still no relation between age leaflets and furanocoumarin concentration.

Reviewer 2: I also do not get the remark that the A5 leaf was apparently discarded?

Authors:   Regarding the A5 leaflet, there was a misunderstanding. The A5 leaflet was not removed from the plant, but excluded from the sampling procedure. A1 was harvested at T0, then A2 at T1, A3 at T2 and A4 at T3. 

Reviewer 2: The english in the new texts needs to be corrected, it is not as good as the rest.

Authors:    The first version of the paper has been extensively reviewed by two editors of American Journal of Expert (Certificate Verification Key: B267-D120-1AE2-524D-7A3E). However, the modifications made during the reviewing process have not been reviewed which resulted in several mistakes. We apologies for this and made extensive editing with native english speakers in order to correct all (or most) of them.

Reviewer 3 Report

The manuscript has improved and is scientifically sound. I would accept this manuscript to publish in present form.

Author Response

Reviewer 3:The manuscript has improved and is scientifically sound. I would accept this manuscript to publish in present form.

Authors:  We thanks reviewer 3 for the time spent on this paper and the remarks he addressed to improve it. We are pleased that the actual version is considered acceptable for publication.

Reviewer 4 Report

The authors added the required information.

I believe this sentence L.90-91 should be revised:

Parsnip plants were submitted to ozone treatments (i.e. 0, 80, 120, 150 and 250 ppb, 4 plants for each concentration) for 6h/day for 3 days.

Please also be consisten with the abbreviation of ultra-high liquid chromatography, I would suggest to stick to UHPLC-MS

Author Response

Authors: We thanks reviewer 4 for the time spent on this paper and the remarks he addressed to improve it.

Reviewer 4: The authors added the required information.

I believe this sentence L.90-91 should be revised:

Parsnip plants were submitted to ozone treatments (i.e. 0, 80, 120, 150 and 250 ppb, 4 plants for each concentration) for 6h/day for 3 days.

Authors:  Modification has been done

Reviewer 4: Please also be consisten with the abbreviation of ultra-high liquid chromatography, I would suggest to stick to UHPLC-MS

Authors: Modifications have been done